

# Association between *EPHA5* methylation status in peripheral blood leukocytes and the risk and prognosis of gastric cancer

Xu Han, Tianyu Liu, Jiabao Zhai, Chang Liu, Wanyu Wang, Chuang Nie, Qi Wang, Xiaojie Zhu, Haibo Zhou and Wenjing Tian

Department of Epidemiology, College of Public Health, Harbin Medical University, Harbin, China

Corresponding author
Wenjing Tian, twj8267@sina.com

## ABSTRACT

**Purpose:** Altered DNA methylation, genetic alterations, and environmental factors are involved in tumorigenesis. As a tumor suppressor gene, abnormal *EPHA5* methylation was found in gastric cancer (GC) tissues and was linked to the initiation, progression and prognosis of GC. In this study, the *EPHA5* methylation level in peripheral blood leukocytes (PBLs) was detected to explore its relationship with GC risk and prognosis.

**Methods:** A total of 366 GC cases and 374 controls were selected as the subjects of this study to collect their environmental factors, and the *EPHA5* methylation status was detected through the methylation-sensitive high-resolution melting method. Logistic regression analysis was utilized to evaluate the associations among *EPHA5* methylation, environmental factors and GC risk. Meanwhile, the propensity score (PS) was used to adjust the imbalance of some independent variables.

**Results:** After PS adjustment, *EPHA5* Pm (positive methylation) was more likely to increase the GC risk than *EPHA5* Nm (negative methylation) ($OR^b$ = 1.827, 95% CI [1.202–2.777], $P$ = 0.005). *EPHA5* Pm had a more significant association with GC risk in the elderly ($OR^a$ = 2.785, 95% CI [1.563–4.961], $P$ = 0.001) and *H. pylori*-negative groups ($OR^a$ = 2.758, 95% CI [1.369–5.555], $P$ = 0.005). Moreover, the combined effects of *EPHA5* Pm and *H. pylori* infection ($OR_c^a$ = 3.543, 95% CI [2.233–5.621], $P$ < 0.001), consumption of alcohol ($OR_c^a$ = 2.893, 95% CI [1.844–4.539], $P$ < 0.001), and salty food intake ($OR_c^a$ = 4.018, 95% CI [2.538–6.362], $P$ < 0.001) on increasing the GC risk were observed. In addition, no convincing association was found between *EPHA5* Pm and the GC prognosis.

**Conclusions:** *EPHA5* methylation in PBLs and its combined effects with environmental risk factors are related to the GC risk.

## INTRODUCTION

Despite its reduced incidence, gastric cancer (GC) ranks fifth among diagnosed malignant tumors and remains the third leading cause of cancer death (*Bray et al., 2018*). GC is asymptomatic in the early stages and most GC cases are diagnosed as distant metastasis at an advanced stage. Once GC progresses to the advanced stage, it is largely incurable and

has a dismal 5-year survival rate (*Ke et al., 2020*). Hence, it is crucial to find effective biomarkers for screening early GC.

GC is a multifactorial disease, and it is well known that genetic alterations, epigenetic alterations and environmental factors contribute to its etiology (*Liabeuf et al., 2022*). Among the oncogenic alterations caused by epigenetic modifications, alterations driven by DNA methylation have been studied most profoundly (*Huang et al., 2022*). It has been firmly established that the activation of tumor suppressor genes could be regulated through DNA hypermethylation of CpG islands (*Li et al., 2022*), which influence tumor progression, the clinical course and the patient's prognosis (*Xu et al., 2017*).

Eph receptors, the largest subfamily of receptor tyrosine kinases, have been implicated in the processes of cellular location, adhesion, migration and differentiation (*Zhu, Li & Mao, 2022*). As a type of Eph receptor, *EPHA5* has been considered an anticancer gene, and it was gradually determined that its altered expression and mutation plays an essential role in the initiation and development of a wide variety of tumors (*Sepulveda et al., 2016*; *Staquicini et al., 2015*). Studies have pointed out that increased *EHPA5* methylation and its corresponding decreased expression are found in GC tissues compared to normal stomach tissues, suggesting that *EPHA5* methylation may promote malignant transformation and neoplastic progression (*Sepulveda et al., 2016*).

DNA in tissue and blood cells contains the frequently variable information of DNA methylation, which has been used as a phenotypic marker to forecast latent cancer risk (*Moore et al., 2008*). Compared with tissue, dysregulated DNA methylation in the peripheral blood is more useful as a biomarker because it is easier to access and causes less damage to patients than tissue biopsies (*Wang et al., 2010*). Therefore, GC patients and controls were recruited to conduct the current case-control study to ascertain the effect of *EPHA5* methylation status in peripheral blood leukocytes (PBLs), environmental factors and their combined actions on GC. Moreover, a follow-up study was subsequently carried out among GC patients to identify whether the *EPHA5* methylation status correlated with the GC prognosis.

## MATERIALS AND METHODS

### Research samples

This is a hospital-based case-control study with 366 cases and 374 controls, which excluded the subjects with a history of cancer, gastrointestinal disease and mental illness. The cases were originated from newly and pathologically diagnosed primary GC patients in the Third Affiliated Hospital of Harbin Medical University from 2010 to 2012. All cases were followed up for 5 years, and finally 345 of them were collected complete prognosis data. The controls were comprised of the ophthalmology and orthopedics patients from the Second Affiliated Hospital of Harbin Medical University, neurology patients of the Fourth Affiliated Hospital of Harbin Medical University and healthy people who took part in the physical examination at the Center for Disease Control and Prevention of Xiangfang District in Harbin. After acquiring written informed consent, all cases and controls were accepted a face-to-face survey to finish a unified questionnaire including demographic data, eating and lifestyle habits, history of disease, and clinical data

of patients. The clinicopathological information was extracted from the electronic medical record system. The overall response rate for cases and controls were approximately 90%. Meanwhile, almost 5 ml blood samples were taken from each subject. This study was approved by the Human Research and Ethics Committee of Harbin Medical University.

## Serologic tests for *Helicobacter pylori*

Serologic tests were utilized to measure the concentration levels of specific immunoglobulin G antibodies against *H. pylori* by ELISA. *H. pylori* positivity was defined as a concentration greater than 12 units/ml, *H. pylori* suspicion was 8–12 units/ml, and *H. pylori* negativity was less than 8 units/ml.

## DNA extraction and bisulfite conversion

A QIAamp DNA Blood Mini Kit (Qiagen, Hilden, Germany) was used to extract genomic DNA from the PBLs, and then the DNA concentration was detected by Nanodrop 2000c (Thermo Scientific, Waltham, MA, USA). An EpiTect Fast DNA Bisulfite Kit (Qiagen, Hilden, Germany) was used to modify the extracted DNA, and the bisulfite-modified DNA was kept at −20 °C.

## Detection of DNA methylation

Primer Premier 5.0 software was implemented to design the *EPHA5* primer pairs, and the bisulfite-modified *EPHA5* primer pairs were 5′-TAAGCGGTATGGGTGTTT-3′ (forward) and 5′-CCTACTATCCCTCACAAACTA-3′ (reverse). The methylation level of *EPHA5* was detected through the methylation-sensitive high-resolution melting method, which was performed on Gene Scanning software (version 2.0) equipped with a LightCycler 480 (Roche Applied Science, Mannheim, Germany). The 5 μl reaction mixture included 2.5 μl Light Cycler 480 High-Resolution Melting Master Mix (Roche, Basel, Switzerland), 1.2 μl PCR-grade water, 0.6 μl $MgCl_2$, 0.5 μl sodium bisulfite-modified DNA template, and 0.1 μl of every forward and reverse primer. A series of standards with 100%, 10%, 5%, 2%, 1%, 0.5%, 0.2% and 0% methylated DNA was constituted by mixing various ratios of bisulfite-modified 100% and 0% human whole genomic DNA (Zymo Research, Irvine, CA, USA). Normalized melting curves and melting peaks of the MS-HRM assay for *EPHA5* are shown in Figs. S1 and S2. The *EPHA5* methylation status in the samples was determined through comparison with the standard curves. Then, 0.2% methylated DNA was selected as the cutoff value of *EPHA5*. Samples with methylated DNA levels higher than 0.2% were assigned to the positive methylation (PM) group, and samples with methylated DNA levels lower than 0.2% were assigned to the negative methylation (NM) group. Instead of a DNA template, DNA-free distilled water was added to the reaction system as a negative control, and a duplicate trial was required if ambiguous results appeared.

## Statistical analysis

The multiple interpolation method was used to impute the missing values. The imbalance of some independent variables was adjusted through the propensity score (PS). Categorical variables were calculated by the chi-square ($\chi^2$) test. Logistic regression analysis was

utilized to explore the relationships among the *EPHA5* methylation level, environmental factors and the risk of GC by computing the odds ratios and the relevant 95% confidence intervals. Crossover analysis was applied to evaluate the associative effects of *EPHA5* methylation and environmental factors on the risk of GC. Multivariate logistic regression analysis with a product-term coefficient was adopted to appraise the interactions of the methylation of *EPHA5* and environmental factors on GC risk. The survival curve of cases was depicted *via* the Kaplan-Meier method, and the log-rank test was performed to compare group differences. Univariate and multivariate Cox regression analyses were used to estimate hazard ratios and 95% confidence intervals to present the effects of *EPHA5* methylation and clinical characteristics on the prognosis of GC patients. Statistical analyses were executed by using SPSS version 23.0. Meanwhile, the R for Windows 3.5.0 and PS matching 3.04 software packages were used to perform the PS matching. Differences were deemed significant when the value of *P* was less than 0.05. Bonferroni correction was performed in the stratified analysis.

## RESULTS

### Demographic characteristics of study samples

In the present study, 366 cases and 374 controls were selected as the study samples, and their basic demographic characteristics are listed in Table 1. There were no significant differences in the distribution of sex ($P = 0.755$) and age ($P = 0.292$) between the cases and controls. The distribution of body mass index (BMI) between the case and control groups was significantly different ($P < 0.001$). Compared with the control group, there was a higher proportion of subjects with monthly income ≥150 dollars (64.2% *vs* 54.8%, $P = 0.012$) and a family history of GC (13.7% *vs* 2.9%, $P < 0.001$) in the case group.

### Associations of environmental factors and GC risk

The relationship of each environmental factor with the risk of GC was assessed by logistic regression analysis, as summarized in Table S1. Multivariate analysis of environmental factors with significant differences was conducted by the backward conditional selection method. Ultimately, a total of 13 environmental factors were selected for the regression model and are shown in Table S2. The results revealed that *H. pylori* infection, a high-salt diet, consuming freshwater fish, dairy product intake, eating fried food, alcohol consumption and eating overnight food significantly increased the GC risk ($P < 0.05$). Meanwhile, a regular diet, green vegetable intake, garlic intake, beef and mutton intake, refrigerated food consumption, and tap and mineral water consumption significantly reduced the GC risk ($P < 0.05$).

### Association of *EPHA5* methylation status and GC risk

The data presented in Table 2 illustrate that the percentages of *EPHA5* methylation were 49.2% and 32.9% in the case and control groups, respectively. After adjusting for all variables in the regression model, *EPHA5* Pm had a statistically significant correlation with the GC risk (OR[a] = 1.935, 95% CI [1.269–2.949], $P = 0.002$). After adjusting for the PS of all

**Table 1 The basic demographic characteristics of the subjects.**

| Variable | | Cases (%) $n = 366$ | Controls (%) $n = 374$ | $P$ |
|---|---|---|---|---|
| Sex | Male | 274 (74.9) | 284 (75.9) | 0.755 |
| | Female | 92 (25.1) | 90 (24.1) | |
| Age (mean ± SD) | | 58.19 ± 11.12 | 59.03 ± 10.44 | 0.292 |
| | <60 | 196 (53.6) | 187 (50.0) | 0.323 |
| | ≥60 | 170 (46.4) | 187 (50.0) | |
| BMI (kg/m²) | <23.00 | 222 (60.7) | 153 (40.9) | <0.001 |
| | ≥23.00 | 144 (39.3) | 221 (59.1) | |
| Monthly income (Dollars/*Per capita*) | <150 | 131 (35.8) | 169 (45.2) | 0.012 |
| | ≥150 | 235 (64.2) | 205 (54.8) | |
| Family history of GC | No | 316 (86.3) | 363 (97.1) | <0.001 |
| | Yes | 50 (13.7) | 11 (2.9) | |

Notes:
The basic demographic characteristics include sex, age, body mass index (BMI), monthly income and a family history of gastric cancer.
BMI, body mass index; GC, gastric cancer.
Differences in sex, age, BMI, monthly income and family history of GC between cases and controls were compared by the $\chi^2$ test.

variables, the result was identical: *EPHA5* Pm significantly increased the GC risk (OR$^b$ = 1.827, 95% CI [1.202–2.777], $P$ = 0.005).

## Stratified analysis of association between *EPHA5* methylation status and GC risk

Age-stratified analysis suggested that a significant association between *EPHA5* Pm and the GC risk was found in the elderly group (≥60 years, OR$^a$ = 2.785, 95% CI [1.563–4.961], $P$ = 0.001). However, no significant association was observed in the younger group (<60 years, Table 3). Stratified analysis by *H. pylori* infection status illustrated that *EPHA5* Pm could increase the risk of GC in the *H. pylori*-negative group (OR$^a$ = 2.758, 95% CI [1.369–5.555], $P$ = 0.005). However, a relationship between *EPHA5* Pm and GC risk was not detected in the *H. pylori*-positive group (Table 3).

## Associations of *EPHA5* methylation status and environmental factors

As shown in Table S3, vegetable and chicken intake both reduced the *EPHA5* methylation level (OR$^b$ = 0.446, 95% CI [0.289–0.687], $P$ < 0.001 and OR$^b$ = 0.627, 95% CI [0.406–0.967], $P$ = 0.035, respectively). Meanwhile, smoking could enhance the *EPHA5* methylation level (OR$^b$ = 1.432, 95% CI [1.058–1.939], $P$ = 0.020).

## Effects of the combination and interaction between *EPHA5* methylation status and environmental factors influence on GC risk

Combined effects were found between *EPHA5* methylation and *H. pylori* infection (OR$_c^a$ = 3.543, 95% CI [2.233–5.621], $P$ < 0.001), consumption of alcohol (OR$_c^a$ = 2.893, 95% CI [1.844–4.539], $P$ < 0.001) and salty food intake (OR$_c^a$ = 4.018, 95% CI [2.538–6.362], $P$ < 0.001), which could affect the GC risk and are listed in Table S4.

**Table 2 Association between methylation status of *EPHA5* and GC risk.**

| Methylation status | | Case (%) | Control (%) | Crude OR (95% CI) | P | OR[a] (95% CI) | P | OR[b] (95% CI) | P |
|---|---|---|---|---|---|---|---|---|---|
| *EPHA5* | Nm | 186 (50.8) | 251 (67.1) | 1.000 | | 1.000 | | 1.000 | |
| | Pm | 180 (49.2) | 123 (32.9) | 1.975 [1.467–2.659] | <0.001 | 1.935 [1.269–2.949] | 0.002 | 1.827 [1.202–2.777] | 0.005 |

Notes:
*EPHA5* Pm has statistically significant correlation with GC risk.
Nm, negative methylation; Pm, positive methylation; CI, confidence interval; OR, odds ratio.
OR was calculated by Logistic regression analysis.
[a] Adjusted for age, sex, BMI, monthly income, family history of GC, alcohol consumption, bean products, beef and mutton, regular diet, eating speed, egg, food left overnight, freshwater fish, fried food, garlic, green vegetables, refrigerated food, salted food, water, dairy products, *H. pylori* infection, smoking of the regression model.
[b] Adjusted for propensity score of all variables.

**Table 3 Association between methylation status of *EPHA5* and risk of GC by stratified analysis.**

| Methylation status | | <60 years | | | | ≥60 years | | | |
|---|---|---|---|---|---|---|---|---|---|
| | | Case (%) | Control (%) | OR (95% CI) | P | Case (%) | Control (%) | OR (95% CI) | P |
| *EPHA5* | Nm | 115 (58.7) | 129 (69.0) | 1.000 | | 71 (41.8) | 122 (65.2) | 1.000 | |
| | Pm | 81 (41.3) | 58 (31.0) | 0.918 [0.464–1.816] | 0.805 | 99 (58.2) | 65 (34.8) | 2.785 [1.563–4.961] | 0.001 |
| Methylation status | | *H. pylori* negative | | | | *H. pylori* positive | | | |
| *EPHA5* | Nm | 62 (45.9) | 133 (68.9) | 1.000 | | 124 (53.7) | 118 (65.2) | 1.000 | |
| | Pm | 73 (54.1) | 60 (31.1) | 2.758 [1.369–5.555] | 0.005 | 107 (46.3) | 63 (34.8) | 1.516 [0.860–2.672] | 0.151 |

Notes:
Significant associations between *EPHA5* Pm and GC risk were found in the elderly and *H. pylori*-negative groups.
Nm, negative methylation; Pm, positive methylation; CI, confidence interval; OR, odds ratio.
OR was calculated by Logistic regression analysis and adjusted for propensity score of all variables except stratified factors. Bonferroni correction with $P < 0.025$ was considered to be statistically significant.

In addition, the interaction between *EPHA5* methylation and a regular diet was found to decrease the GC risk (OR$_i^a$ = 0.338, 95% CI [0.142–0.803], $P = 0.014$).

## Demographic characteristics of cases

A total of 345 GC patients were selected to participate in the 5-year follow-up study. The demographic characteristics and their associations with the prognosis of GC patients are shown in Table S5. The results confirmed that none of the demographic characteristics were significantly correlated with the prognosis of GC. However, as common confounding factors, age, sex and BMI were adjusted to control for confounding bias when the effects of clinical characteristics on the GC prognosis were estimated. Some conclusions could be drawn from the multivariate Cox analysis that tumor size, differentiation, tumor–node–metastasis (TNM) stage, and the levels of both carbohydrate antigen 19–9 (CA 19–9) and carcinoembryonic antigen (CEA) could influence the GC prognosis (all $P < 0.05$, Table 4).

Analysis through backward conditional selection revealed that tumor size had a statistically significant correlation with the GC prognosis (HR = 1.589, 95% CI [1.158–2.181], $P = 0.004$). In addition, TNM stages III and IV were both strongly associated with a worse GC prognosis (HR = 3.227, 95% CI [1.422–7.323], $P = 0.005$ and HR = 6.473, 95% CI [2.995–13.992], $P < 0.001$, respectively, Table S6).

**Table 4 Association between clinical characteristics and prognosis of GC.**

| Clinical characteristics | | Case (%) | HR (95% CI) | P | HR[a] (95% CI) | P |
|---|---|---|---|---|---|---|
| Tumor site | Distal stomach | 214 (62.0) | 1.000 | | 1.000 | |
| | Others | 131 (38.0) | 1.323 [0.967–1.808] | 0.080 | 1.341 [0.979–1.837] | 0.067 |
| Tumor size | <5 cm | 169 (49.0) | 1.000 | | 1.000 | |
| | ≥5 cm | 176 (51.0) | 2.209 [1.608–3.034] | <0.001 | 2.214 [1.608–3.049] | <0.001 |
| Pathological typing | Polypoid type | 27 (7.8) | 1.000 | | 1.000 | |
| | Ulcer type | 53 (15.4) | 0.680 [0.311–1.486] | 0.333 | 0.708 [0.321–1.561] | 0.391 |
| | Infiltrating ulcer type | 197 (57.1) | 1.268 [0.693–2.319] | 0.441 | 1.301 [0.709–2.388] | 0.395 |
| | Infiltrating type | 61 (17.7) | 1.804 [0.918–3.545] | 0.087 | 1.849 [0.936–3.655] | 0.077 |
| | Other type | 7 (2.0) | 0.422 [0.063–2.833] | 0.373 | 0.445 [0.065–3.035] | 0.407 |
| Histological type | adenocarcinoma | 186 (53.9) | 1.000 | | 1.000 | |
| | Particular types carcinoma | 101 (29.3) | 0.641 [0.444–0.927] | 0.018 | 0.653 [0.451–0.947] | 0.025 |
| | mixed carcinoma | 58 (16.8) | 0.814 [0.531–1.249] | 0.346 | 0.797 [0.517–1.230] | 0.306 |
| Differentiation | Low | 208 (60.3) | 1.000 | | 1.000 | |
| | Middle to High | 137 (39.7) | 0.681 [0.481–0.964] | 0.030 | 0.680 [0.479–0.964] | 0.031 |
| TNM stage | I | 50 (14.5) | 1.000 | | 1.000 | |
| | II | 18 (5.2) | 2.115 [0.607–7.366] | 0.239 | 2.068 [0.588–7.273] | 0.257 |
| | III | 77 (22.3) | 3.242 [1.294–8.126] | 0.012 | 3.162 [1.255–7.969] | 0.015 |
| | IV | 200 (58.0) | 8.074 [3.386–19.253] | <0.001 | 7.832 [3.277–18.720] | <0.001 |
| CA19–9 | <37 u/ml | 276 (80.0) | 1.000 | | 1.000 | |
| | ≥37 u/ml | 69 (20.0) | 1.554 [1.069–2.259] | 0.021 | 1.621 [1.106–2.374] | 0.013 |
| CEA | <5 ng/ml | 269 (78.0) | 1.000 | | 1.000 | |
| | ≥5 ng/ml | 76 (22.0) | 1.582 [1.106–2.262] | 0.012 | 1.646 [1.149–2.359] | 0.007 |

Notes:
Multivariate Cox analysis that tumor size, differentiation, tumor–node–metastasis (TNM) stage, the level of both carbohydrate antigen 19–9 (CA 19–9) and carcinoembryonic antigen (CEA) could influence on GC prognosis.
CI, confidence interval; HR, hazard ratio.
HR was calculated by Cox regression analysis.
[a] Adjusted for age, sex, BMI.

**Table 5 Association between methylation status of EPHA5 and GC prognosis.**

| Methylation status | | Case (%) | HR (95% CI) | P | HR[a] (95% CI) | P | HR[b] (95% CI) | P |
|---|---|---|---|---|---|---|---|---|
| *EPHA5* | Nm | 180 (52.2) | 1.000 | | 1.000 | | 1.000 | |
| | Pm | 165 (47.8) | 0.971 [0.724–1.302] | 0.845 | 0.896 [0.661–1.213] | 0.477 | 0.940 [0.692–1.278] | 0.694 |

Notes:
No significant correlation was found between *EPHA5* methylation and GC prognosis.
Nm, negative methylation; Pm, positive methylation; CI, confidence interval; HR, hazard ratio.
HR was calculated by Cox regression analysis.
[a] Adjusted for age, sex, BMI, tumor size, TNM stage.
[b] Adjusted for propensity score of all variables.

## Association of *EPHA5* methylation status and the prognosis of GC

As the results in Table 5 indicate, whether multivariate or PS adjustment was used to examine the relationship between *EPHA5* methylation and GC prognosis, no significant correlation was found. Meanwhile, the relationship between *EPHA5* methylation and GC prognosis was depicted *via* Kaplan–Meier survival curves in Fig. 1.

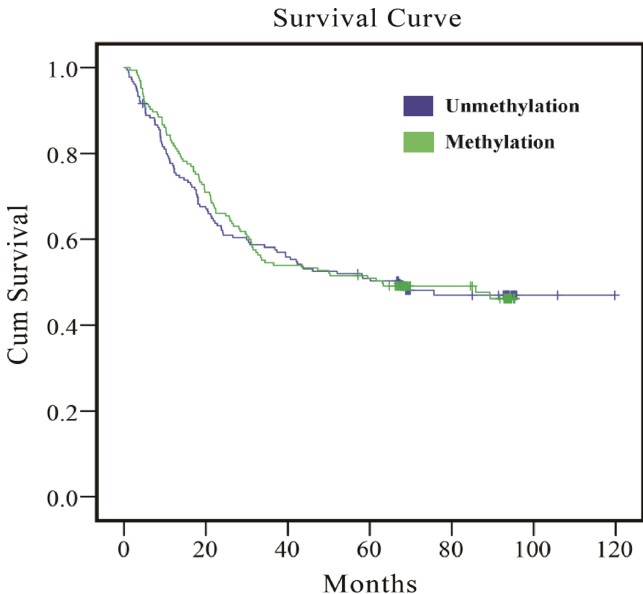

**Figure 1 Survival curves of the association between methylation status of *EPHA5* and GC prognosis.** The purple line indicates the unmethylation group, and the green line indicates the methylation group.

## Stratified analysis of the association between *EPHA5* methylation status and GC prognosis

The analysis was stratified by sex, age, *H. pylori* infection, TNM stage, and tumor size, and the results demonstrated that *EPHA5* Pm was marginally associated with the GC prognosis in the younger subgroup and in the tumor larger than 5 cm subgroup, with borderline statistical significance (HR[b] = 0.645, 95% CI [0.409–1.016], $P$ = 0.059 and HR[b] = 0.689, 95% CI [0.462–1.025], $P$ = 0.066, Tables S7 and S8). The interpretation of this result needs to be based on a larger sample size. In addition, there was no statistical correlation found between *EPHA5* Pm and the GC prognosis in the sex, *H. pylori* infection and TNM stage subgroups (Tables S9–S11).

## DISCUSSION

GC is a sustained multistep process that involves accumulated genetic damage, altered epigenetic signatures, and exposure to environmental factors. An increasing number of studies have confirmed that epigenetic alterations can mediate the effects of environmental factors on disease (*Agache et al., 2020*). As the most typical epigenetic alteration, DNA hypermethylation is closely linked to improper transcriptional silencing and the functional loss of genes to accelerate the growth of tumor cells and promote tumorigenesis and development (*Abudurexiti et al., 2020*). To screen for early-stage tumors, methylation-based biomarkers detected from tissue samples were highlighted in the recent long-term study (*Ibrahim et al., 2019*). However, PBLs are the optimum samples to explore methylation-based biomarkers, because of their easy access. A great variation in methylation status has been observed in the PBLs between malignant and nonmalignant individuals (*Wang et al., 2010*). Consequently, it is important to determine

the methylation status of genes in PBLs to analyze their correlation with the GC risk and prognosis.

Previous researchers applied bisulfite next-generation sequencing and The Cancer Genome Atlas to seek biomarkers for GC, and DNA hypermethylation and reduced RNA expression levels of *EPHA5* were detected in GC tissues compared with the corresponding non-gastric cancer tissues from the same case (*Sepulveda et al., 2016*). Our data showed the analogous conclusion from PBLs that *EPHA5* Pm had a significant correlation with GC risk.

Stratified analysis based on age and *H. pylori* infection was performed in this study. *Cruickshanks et al. (2013)* suggested that with age, the evolution from global DNA hypomethylation to DNA hypermethylation of CpGs could be verified in mammalian cells, which was likely induced by the differential expression of DNMTs. It is noteworthy that alterations in DNMT expression can lead to epigenetic instability and then stimulate the occurrence of age-related disease (*Sen et al., 2016*). This may provide an explanation for the result of the age-stratified analysis that individuals with *EPHA5* Pm are more likely to be diagnosed with GC at older ages. Moreover, some studies have reported that *H. pylori* infection could increase DNA methylation (*Lu et al., 2021*), which could explain the result in this study that *EPHA5* methylation occurred in both cases and controls in the *H. pylori*-positive group. Therefore, a relationship between *EPHA5* Pm and GC risk was not found in the *H. pylori*-positive group, but the reason why a relationship was found in the *H. pylori*-negative group may need additional stratification analysis.

A combined effect of *EPHA5* Pm and *H. pylori* infection on increasing the GC risk was observed in the current study. Reportedly, *H. pylori* infection promotes the accumulation of promoter hypermethylation of a number of genes that are closely relevant to GC (*Zhao et al., 2020*). The aberrant DNA methylation caused by *H. pylori* infection could be partly reversed after *H. pylori* eradication (*Shin et al., 2013*). In addition, numerous studies have shown that *H. pylori*-induced chronic inflammation activates macrophages, induces DNMT1 transcription, and then causes altered DNA methylation (*Chiba, Marusawa & Ushijima, 2012*). A combined effect between *EPHA5* Pm and alcohol consumption on GC risk was also found in this study. It has been shown that heightened plasma homocysteine levels are detected in individuals with alcohol dependence, resulting in higher levels of gene-specific DNA methylation in peripheral blood cells (*Bleich et al., 2006*). We also found a combined effect between *EPHA5* Pm and a high-salt diet on GC risk. Some researchers posited that the consumption of a low-salt diet could cause DNA hypomethylation in a gene-specific manner by increasing the synthesis of aldosterone. Switching from a low-salt diet to a high-salt diet reduced the synthesis of aldosterone, and the DNA hypomethylation status of certain genes was reversed to a DNA hypermethylation status (*Takeda et al., 2018*).

Relevant studies have suggested that epigenetic alterations might be caused by diverse types of environmental factors, which could be conducive to cancer development (*Wiese & Bannister, 2020*). Previous studies have reported that vegetables contain many types of vitamins, and their consumption is inversely associated with gene-specific DNA methylation (*Fujii et al., 2019*). Experimental studies have elucidated that vitamin C could

induce DNA demethylation by enhancing the activity of ten-eleven translocation enzymes (*Chen et al., 2013*; *Yin et al., 2013*). In addition, vitamin E plays an antioxidant role in the DNA methyltransferase pathways, so that the level of DNA methylation is reduced (*Remely et al., 2017*). Although few studies have reported the mechanism linking β-carotene, a dietary form of vitamin A, and DNA methylation, a reduction in the level of gene-specific DNA methylation after β-carotene intake was detected (*Fujii et al., 2019*). In addition, the results obtained from a previous study showed that long-term exposure to tobacco increased the expression of DNA methyltransferase 3B (DNMT3B) and recruited DNMT3B to the promoter regions of certain genes, promoting the hypermethylation of these genes and their carcinogenic properties (*Shiah et al., 2020*). In line with these findings, negative and positive associations were identified between *EPHA5* methylation and vegetable intake and smoking, respectively, in our study.

Convincing evidence provided from epidemiological and mechanistic studies showed that dietary habits and lifestyles are involved in the development of the great majority of neoplasms (*Brouwer et al., 2021*). Many environmental factors may increase the GC risk, including a high-salt diet, consuming freshwater fish, eating fried food, dairy product consumption, alcohol consumption and eating overnight food (*Huang et al., 2020*; *Karimi et al., 2014*). In addition, accumulating evidence has shown that a regular diet, eating refrigerated food, vegetable intake, and the consumption of garlic and tap and mineral water were protective factors against GC, and our results are consistent with these previous findings (*Cai, Zheng & Zhang, 2003*; *Howson, Hiyama & Wynder, 1986*; *Kim et al., 2013*; *Li et al., 2019*; *Sitarz et al., 2018*). However, the results for beef and mutton in this study were different from previous findings. Previous findings showed that beef and mutton could increase the GC risk (*Collatuzzo et al., 2022*), but a correlation between beef and mutton intake and a reduced GC risk was observed in the current study. In light of the differences between these two viewpoints, our understanding is that red meat cooking methods vary and the formation of different chemicals depends on the different cooking temperatures and times (*Di Maso et al., 2013*; *Gharehbeglou & Jafari, 2019*). For example, the GC risk was increased when red meat was cooked by boiling or stewing. An association between roasting/grilling and frying/pan frying red meat and a GC risk was not found in previous studies (*Di Maso et al., 2013*).

A follow-up study based on 345 GC cases was launched to analyze the effect of the *EPHA5* methylation status on GC prognosis. Our data validated that the tumor size and TNM classification were significantly correlated with the GC prognosis, which is consistent with previous observations (*Wei et al., 2022*). Previous reports have shown that *EPHA5* methylation is associated with a poor prostate cancer prognosis (*Li et al., 2015*). Furthermore, decreased expression of the *EPHA5* gene is seen after hypermethylation, which was associated with a poor breast cancer prognosis (*Madden et al., 2013*). However, a relationship between *EPHA5* Pm and GC prognosis was not found in this study. To clarify further whether there is an association between *EPHA5* Pm and the GC prognosis, a mechanistic study or an epidemiological study with a larger sample size needs to be performed in the future.

Some limitations of the present article should be considered in future studies. One obstacle was that as a case-control study, the causality of the gene methylation status and GC risk could not be proven. It is thus necessary to conduct a prospective cohort study. Another drawback was that the emergence of recall bias was inevitable when we collected the environmental factors of the subjects, although PS analysis was applied to balance the discrepancies at baseline. Additionally, only considering the impact of food intake on GC was incomplete, because it is known that the cooking method could influence the association between food and GC risk.

## CONCLUSIONS

*EPHA5* methylation in peripheral blood leukocytes was associated with the risk of GC and it is a potential methylation-based biomarker to predict, screen and diagnose GC. Furthermore, the combined effects of *EPHA5* methylation and *H. pylori* infection, consumption of alcohol and salty food intake might increase the risk of GC.

## ACKNOWLEDGEMENTS

We would like to thank all the patients who participated in this study.

### Funding
This study was supported by the National Natural Science Foundation of China (No. 81573219). The funders had no role in study design, data collection and analysis, decision to publish, or preparation of the manuscript.

### Grant Disclosures
The following grant information was disclosed by the authors:
National Natural Science Foundation of China: 81573219.

### Competing Interests
The authors declare that they have no competing interests.

### Author Contributions
- Xu Han conceived and designed the experiments, performed the experiments, analyzed the data, authored or reviewed drafts of the article, and approved the final draft.
- Tianyu Liu performed the experiments, authored or reviewed drafts of the article, and approved the final draft.
- Jiabao Zhai performed the experiments, authored or reviewed drafts of the article, and approved the final draft.
- Chang Liu analyzed the data, prepared figures and/or tables, and approved the final draft.
- Wanyu Wang analyzed the data, prepared figures and/or tables, and approved the final draft.

- Chuang Nie conceived and designed the experiments, analyzed the data, prepared figures and/or tables, and approved the final draft.
- Qi Wang conceived and designed the experiments, prepared figures and/or tables, and approved the final draft.
- Xiaojie Zhu conceived and designed the experiments, prepared figures and/or tables, and approved the final draft.
- Haibo Zhou conceived and designed the experiments, authored or reviewed drafts of the article, contributed reagents and materials, and approved the final draft.
- Wenjing Tian conceived and designed the experiments, authored or reviewed drafts of the article, and approved the final draft.

## Human Ethics

The following information was supplied relating to ethical approvals (*i.e.*, approving body and any reference numbers):

The Human Research and Ethics Committee of Harbin Medical University approved this study.

## Data Availability

The raw measurements are available in the Supplemental Files.

## Supplemental Information

Supplemental information for this article can be found online at http://dx.doi.org/10.7717/peerj.13774#supplemental-information.

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
