# Peer review of "Association between EPHA5 methylation status in peripheral blood leukocytes and the risk and prognosis of gastric cancer"

_PeerJ, doi:10.7717/peerj.13774_

## Round 0.1 · original submission · Major Revisions

As the authors will realize, the content and quality of the manuscript need to be improved with a major revision. Therefore, I would like to invite the authors to implement the referee comments accordingly.

·

Basic reporting

The paper investigates the association between the methylation status of the tumor suppressor gene EPHA5 in peripheral blood leukocytes and the environmental factors, as well as the risk and prognosis of gastric cancer. Although the topic is interesting and the authors have conducted both rigorous wet-lab experiments and computational analyses, the overall quality of the paper can be improved by incorporating the following recommendations:
- The paper is awkwardly written and difficult to understand, with many grammatical/typo/spelling/style/terminology errors (e.g., genetic alternations). A thorough revision of the entire manuscript is needed by a fluent speaker.
- More recent references can be added.
- Better description of Materials & Methods.

Experimental design

The Materials & Methods are not described sufficiently. The authors should be more explicit; for example, the method and the software used to estimate the Hazard Ratio (HR) from the Kaplan-Meier curves is not explained.

Validity of the findings

The Results are in overall well-supported. However, some points must be explained (e.g., what does RMB stand for?), and less assertive language should be used.

Additional comments

In the Discussion the findings are addressed quite adequately. However, more updated references could be added to further support the results.

Reviewer 2 ·

Basic reporting

Good

Experimental design

Good

Validity of the findings

In the present study, Han et al, showed the prognostic value of EPHA5 in Gastric cancer and evaluated the association of methylation status in peripheral blood leukocytes. Based on my own evaluation following concerns are there to consider-
1. Nm and Pm should be defined for the abbreviation used in the abstract.
2. Author mentioned that 366 GC patients’ samples were collected and used from 2010 to 2012. Is this number patient available from a single institute or from multiple institute?
3. The primer pairs used for the EPHA5 gene does not show any target templates in selected database for Homo sapiens.
4. Author should provide the rationale why EPHA5 methylation in PBL was used for this study?
5. Kaplan-Meier survival curve does not show any survival benefits in Nm or Pm in GC as shown in figure 1 which shows that conclusion is overdrawn.

---

## Round 0.2 · Minor Revisions

The authors successfully addressed both reviewers' comments. However, a very minor change and needs to be made before accepting the manuscript for publication.

·

Basic reporting

Most of my comments have been addressed.
The manuscript has been also revised by a professional editing service.

Experimental design

Satisfactory

Validity of the findings

Good

Reviewer 2 ·

Basic reporting

Good

Experimental design

Good

Validity of the findings

satisfactory

Additional comments

Need to incorporate the Figure 1: survival curve in main text

---

## Round 0.3 · accepted · Accept

The authors implemented the reviewer's comment. The manuscript can now be accepted as it is.